# Health and economic burden of insufficient physical activity in Saudi Arabia

Saleh A. Alqahtani[1,2]*, Reem AlAhmed[3], Mariam M. Hamza[4], Saleh A. Alessy[5,6],
Ada Alqunaibet[7], Amal AlGhammas[8], David Watkins[9], William Msemburi[10],
Fadiah Alkhattabi[11], Sarah Pickersgill[12], Severin Rakic[4], Reem F. Alsukait[13], Christopher
H. Herbst[4], Hazzaa M. Al-Hazzaa[14]

**1** Organ Transplant Center of Excellence, King Faisal Specialist Hospital & Research Center, Riyadh, Saudi Arabia, **2** Division of Gastroenterology & Hepatology, Johns Hopkins University, Baltimore, MD, United States of America, **3** Biostatistics, Epidemiology and Scientific Computing Department, King Faisal Specialist Hospital & Research Center, Riyadh, Saudi Arabia, **4** The World Bank Group, Washington, DC, United States of America, **5** Public Health Department, College of Health Sciences, Saudi Electronic University, Riyadh, Saudi Arabia, **6** Centre for Cancer, Society & Public Health, Faculty of Life Sciences & Medicine, King's College London, London, United Kingdom, **7** Public Health Authority, Riyadh, Saudi Arabia, **8** Academic and Training Affairs Department, King Faisal Specialist Hospital & Research Center, Riyadh, Saudi Arabia, **9** Independent Consultant, Seattle, WA, United States of America, **10** Independent Consultant, Cape Town, South Africa, **11** Department of Pediatrics, King Faisal Specialist Hospital and Research Center, Riyadh, Saudi Arabia, **12** Independent Consultant, Oakland, CA, United States of America, **13** Department of Community Health Sciences, College of Applied Medical Sciences, King Saud University, Riyadh, Saudi Arabia, **14** Lifestyle and Health Research Center, Health Science Research Center, Princess Nourah bint Abdulrahman University, Riyadh, Saudi Arabia

* salalqahtani@kfshrc.edu.sa

**Data Availability Statement:** All input data and software code used in our analysis are available at:

## Abstract

### Background

Insufficient physical activity (PA) was estimated to cause 4.8% of deaths and 2.6% of disability-adjusted life-years (DALYs) due to noncommunicable diseases in Saudi Arabia in 2019. While Saudi Arabia is already achieving great improvements, we predict the health and economic burden of insufficient PA up to 2040 to present a case for policy makers to invest more in the uptake of PA.

### Methods

Using a population health model to estimate avoidable health loss, we identified four causes of health loss related to low PA (cardiovascular diseases, diabetes, breast cancer, and colorectal cancer) and estimated the deaths and DALYs from these causes. We projected the expected disease burden until 2040 under alternative assumptions about future PA levels and trends by using three health scenarios: baseline (no change in 2019 PA levels), intervention (81% of the population achieving sufficient PA levels), and ideal (65% of population: moderate PA, 30%: high PA, and 5%: inactive). We applied an "intrinsic value" approach to estimate the economic impact of each scenario.

https://github.com/Disease-Control-Priorities/PA_KSA.

**Funding:** This work was supported by the King Faisal Specialist Hospital and Research Center and World Bank. Financing for the analysis was provided by the King Faisal Specialist Hospital and Research Center and the Health, Nutrition, and Population Reimbursable Advisory Services Programs between the World Bank and the Ministry of Finance in Saudi Arabia (P172148 and P179873). The content is solely the responsibility of the authors and does not necessarily represent the official views of the King Faisal Specialist Hospital and Research Center or the World Bank, or the governments they represent. The funders had no role in study design, data collection and analysis, decision to publish, or preparation of the manuscript.

**Competing interests:** The authors have declared that no competing interests exist.

## Results

Overall, we estimate that between 2023 and 2040, about 80,000 to 110,000 deaths from all causes and 2.0 million to 2.9 million DALYs could be avoided by increasing PA levels in Saudi Arabia. The average annual economic loss from insufficient PA is valued at 0.49% to 0.68% of the current gross domestic product, with an average of US$5.4 billion to US$7.6 billion annually till 2040. The most avoidable disease burden and economic losses are expected among males and because of ischemic heart disease.

## Conclusions

This study highlights that low PA levels will have considerable health and economic impacts in Saudi Arabia if people remain inactive and do not start following interventions. There is an urgent need to develop innovative programs and policies to encourage PA among all age and sex groups.

## Introduction

Noncommunicable diseases (NCDs) are responsible for about two-thirds of deaths in Saudi Arabia [1] and represent a growing burden to healthcare systems and societies. Trends in NCDs are partly driven by population growth and aging but also by increased exposure to risk factors, such as unhealthy diet, tobacco use, and insufficient physical activity, including sedentary behavior.

Globally, 7.2% and 7.6% of all-cause and cardiovascular disease deaths, respectively, are attributable to physical inactivity [2]. The proportions of NCDs attributable to physical inactivity vary by world region and level of income, with recent worldwide estimates showing that in Saudi Arabia physical inactivity accounts for some of the highest risk for cardiovascular diseases, dementia, and cancer [2]. Insufficient physical activity contributes to nearly 5% of deaths in Saudi Arabia [3]. The country is currently undertaking significant transformation through Vision 2030 [4]. As part of this initiative, the Quality of Life Program aims at getting 40% of the adult population to meet physical activity recommendations by 2030, by engaging in physical activity at least 30 minutes a week [5].

According to the Saudi General Authority of Statistics (GASTAT), this goal has been achieved with 48.2% of the population now partaking in physical activity [6, 7]. Albeit meeting the Quality of Life Program target represents clear progress, it remains below the standard recommendation provided by the World Health Organization and Centers for Disease Control and Prevention, which advocate for 150 minutes of moderate-intensity physical activity and two days of muscle-strengthening activity per week [8, 9]. Therefore, insufficient physical activity can be defined as not meeting the current recommended daily physical activity guidelines of doing at least 150 minutes of moderate-intensity or 75 minutes of vigorous-intensity physical activity per week or any equivalent combination of the two [2]. Under this definition, over half of adults worldwide are not meeting the recommendations for physical activity [10].

The related concepts of the "economic burden of disease" and "cost of inaction" have been around for many years [11], but interest has been renewed due to the dramatic increase in sedentary lifestyles during the pandemic. As economic development continues to transform the labor force regionally towards more digital-based platforms and enclosed environments, physical activity levels are expected to worsen—unless policy action is taken [12].

Studies examining the cost of inaction aim to quantify the cost of not addressing a particular disease, injury, or risk factor. These analyses often have counterfactual arguments, e.g., if disease X had been eliminated, there would have been Y economic benefits (usually valued in local or international currency). These estimates provide insight into the economic benefits of addressing health problems and are useful for agenda-setting, policy formulation and analysis, and advocacy purposes. Estimating the current losses due to physical inactivity will help set priorities to achieve more ambitious goals [4, 5] and help to design a framework to promote a balance between work commitments and physical activity.

Given the high burden caused by physical inactivity in Saudi Arabia, our study investigates the economic impact of health outcomes related to insufficient physical activity. We develop three hypothetical future health scenarios, based on alternative projections of physical activity levels in the population, and estimate the number of deaths, and disability-adjusted life-years (DALYs) that could be avoided as a result of adequate physical activity. These health gains are then translated into economic returns using standard cost-benefit analysis methods, which could inform policy-making decisions at a multi-sectorial level in Saudi Arabia.

## Methods

### Overview

Based on the consensus of the 2019 Global Burden of Disease (GBD) Risk Factors Collaborators [3], and on a previous related dose-response meta-analysis [13], four major causes of health loss were identified, with robust evidence for a causal association with insufficient physical activity: cardiovascular diseases, diabetes, breast cancer, and colorectal cancer [14]. The association is stronger for cardiovascular diseases and diabetes [15] but weaker for cancers [16]. Fig 1 provides a conceptual overview of our analysis.

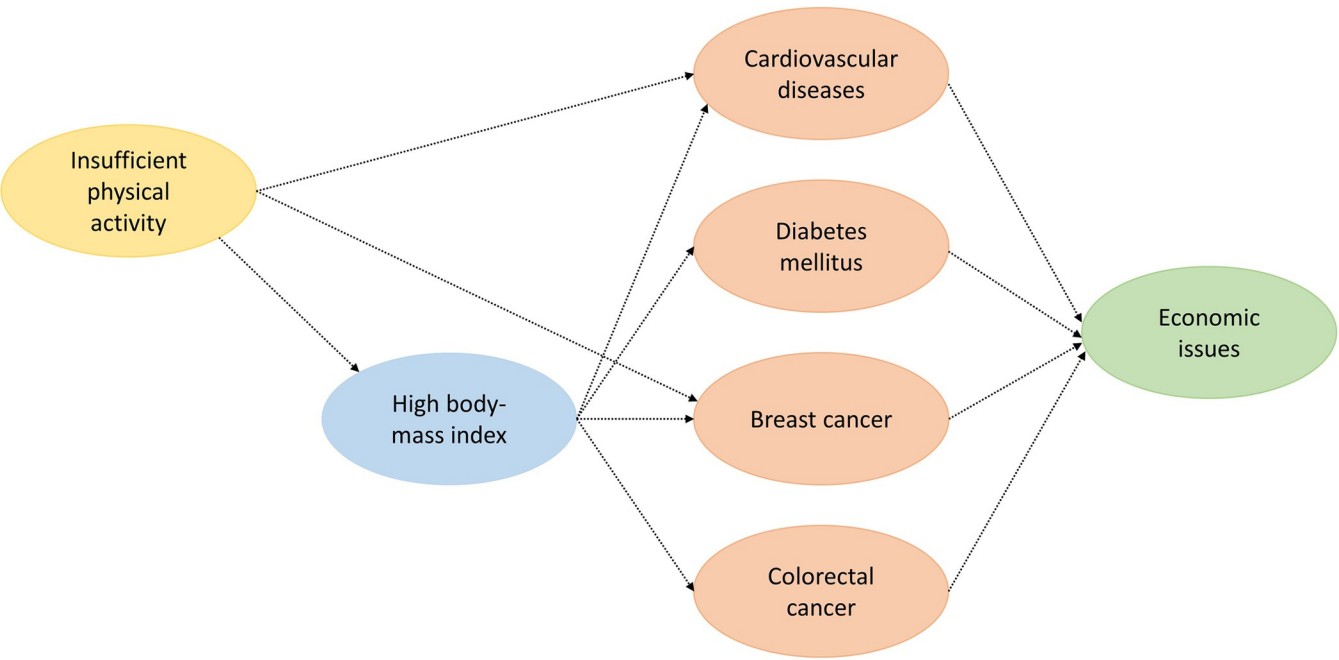

**Fig 1. Conceptual overview of the analysis.**

Common to all approaches that estimate the health and economic impact of diseases and risk factors are two components: (i) estimated "avoidable" disease burden (left side of Fig 1) and then (ii) calculated economic value of that avoidable burden (right side of Fig 1).

## Estimating avoidable health loss

This study adapts a population health model previously developed for NCD prevention and control interventions in different countries [17] to project the expected disease burden in Saudi Arabia under alternative assumptions about future physical activity levels and trends.

We use a cohort-component projection approach to model the evolution of the Saudi population's size and structure over time. We use population estimates starting (2019) and assumed future levels of age- and sex-specific all-cause mortality. All-cause mortality is decomposed into two components: (i) mortality from the four causes amenable to physical activity (Fig 1) and (ii) mortality from other causes. In the two scenarios described below, we model a reduction in all-cause mortality due to reduced mortality from component (i), which itself is due to a shift in the distribution of physical activity in the population. Component (ii) is assumed to remain constant over time.

To model the reduction in cause-specific mortality due increased physical activity, we develop state-transition (Markov) models for each of the four causes shown in Fig 1. Each model includes four states: 'alive without disease X,' 'alive with disease X,' 'dead from disease X,' and 'dead from another cause.' The transition probabilities in the four models are informed by GBD 2019 estimates of age- and sex-specific disease incidence and mortality. The transition probabilities are calibrated using an algorithm described previously for a related set of state-transition models for cardiovascular disease [18]. We assume that the impact of improved physical activity on mortality would be due to a reduction in the incidence of each cause, i.e., the transition from 'alive without disease X' to 'alive with disease X.' Reduced incidence would result in fewer cases of disease and therefore lower mortality levels over time.

To model the reduction in disease-specific incidence, we define risk categories for the population based on the 2019 GBD study's pooled relative risk (RR) estimates for each of the four cause groups [3]. These RRs are specific to increments of 600 metabolic equivalents (MET) minutes per week (e.g., 0, 600, 1200, 1800, etc.). We define the population distribution of physical activity using three discrete categories that correlate well with the dose-response relationship that has been reported in the literature [13]:

1. **Low:** Below 600 MET min/week. To get a proxy risk level for this group, we average the RR values for 0 MET min/week and 600 MET min/week for each age and sex group.

2. **Moderate:** Between 600 MET min/week to 1800 MET min/week. We use age- and sex-specific RR values for 1200 MET min/week as the proxy risk level for this group.

3. **High:** Over 1800 MET min/week. As a conservative measure, we use age- and sex-specific RR values for 1800 MET min/week as the proxy risk level for this group.

Population distributions of current physical activity uptake in Saudi Arabia are obtained from the 2019 GASTAT Household Sports Survey [19].

Integral to estimating avoidable disease burden is to specify the health scenarios that define the bounds of "avoidable" accounting for the different risk categories. We model three such scenarios for Saudi Arabia:

1. **Baseline:** Assuming that physical activity levels reported in the 2019 GASTAT Household Sports Practice Survey [19] remain constant until 2040, with no further progress on increasing physical activity in the coming years. This servs as the reference scenario for the other

two scenarios, wherein age- and sex-specific disease incidence would be constant over time, rather than decreased in relation to increased physical activity.

2. **Intervention:** Assuming that physical activity levels could shift in 2023 to those observed in a high-performing (benchmark) country, reflecting ambitious but realistic changes in population behaviors due to physical activity promotion interventions and maintained until 2040. In this analysis, we define Sweden as the benchmark country, as being one of the better countries to implement these interventions and having the lowest levels of inactive populations [20–22]. The assumption we use for this scenario, based on data from Sweden, is that 81% of the population would achieve sufficient levels of physical activity, defined as the recommended minimum of 150 min/week of moderate physical activity [9, 10, 23].

3. **Ideal:** Assuming that physical activity could shift in 2023 to "ideal levels," reflecting the maximum possible health impact of improved physical activity and maintained until 2040. We define "ideal" as 65% participating in at least 150 min/week of moderate physical activity, another 30% of the population participating in more intense and/or frequent levels of physical activity above the minimum recommended, and the remaining 5% of the population assumed to be inactive for health or aging-related reasons (e.g., spinal cord injury, severe dementia) and unable to achieve at least the 150-minute recommendation.

To calculate avoidable health loss due to insufficient physical activity, the potential impact fraction (PIF) [24] is calculated for each of the four causes of death linked to inadequate physical activity, as well as by age group and sex. Potential impact fraction values are based on the population distribution (P) of physical activity and relative risk (RR) in each risk category (i):

$$PIF = \frac{\sum_i P_i * RR_i - \sum_i \hat{P}_i * RR_i}{\sum_i P_i * RR_i}$$

where $P_i$ is the baseline risk distribution (i.e., the proportion of the population participating in each level of physical activity according to the 2019 GASTAT Household Sports Practice survey data) [19], and $\hat{P}_i$ is the alternative risk distribution in the intervention and ideal scenarios. RR estimates for colorectal cancer, breast cancer, ischemic heart disease, ischemic stroke, and type 2 diabetes mellitus for males and females aged 25–79 years are taken from other GBD studies [3, 13]. The potential impact fraction is applied to the baseline projection of disease-specific incidence and hence mortality as a relative reduction in the former to yield alternative estimates of age-, sex-, and cause-specific deaths. For example, a PIF of 0.10 for a given age/sex/cause group and scenario would imply a 10% reduction in incidence for that group in that scenario as compared to the baseline projection. The difference between baseline, intervention, and ideal scenario projections of disease-specific and all-cause incidence and mortality are reported as 'cases averted' and 'deaths averted' respectively.

We then translated the cases and deaths averted into DALYs averted. Years of life lost averted is calculated as the product of age- and sex-specific deaths averted and the global 'frontier' age- and sex-specific remaining life expectancy as reported in the 2019 GBD 2019. Years lived with disability averted are calculated as the product of cause-specific cases averted and the average disability weights for each cause as reported in the 2019 GBD study [1].

High body mass index is an intermediary risk factor for these disease outcomes. Still, the literature suggests that insufficient physical activity is a risk factor for diseases independent of high body mass index [15], so our analysis used these RRs for inadequate physical activity of the former [13].

### Estimating economic impact

Generally, speaking, there are three approaches to estimating the economist impact of ill health [25]. The first is the "cost-of-illness" approach, which seeks to quantify direct and indirect costs of medical care for particular diseases or injuries, usually at the individual patient level. The second is the "economic growth" approach, which estimates the depletion of labor and capital at the macroeconomic level that is due to particularly diseases or injuries. The third is the "intrinsic value" or "welfare" approach, which uses willingness-to-pay metrics like the value of a statistical life to value improvements in health. The third is the most commonly used approach in benefit-cost analysis and is used in this study.

Specifically, we convert the value of a statistical life into the value of a life-year using methods described previously [24], for Saudi Arabia, the value of a life-year was 2.3 times gross domestic product (GDP) per capita. We then multiply this quantity by the number of DALYs averted in the intervention and ideal scenarios relative to the baseline scenario [24].

## Results

### Potential impact fractions

The PIFs calculated from baseline physical activity levels and RRs illustrate the relative impact that improved physical activity would have on the incidence of the four disease groups featured in our analysis. Fig 2 plots the range of PIF values across the different five-year age groups in the 25–79 years range (y-axis), disaggregated by scenario and cause of death (x-axis). We present results for females and males separately because of the substantial difference in PIFs between the two sexes. These calculations reveal three main findings. First, unsurprisingly, the PIFs are larger for the ideal scenario than for the intervention scenario because of a greater assumed increase in physical activity. Second, PIFs are generally higher for females than males

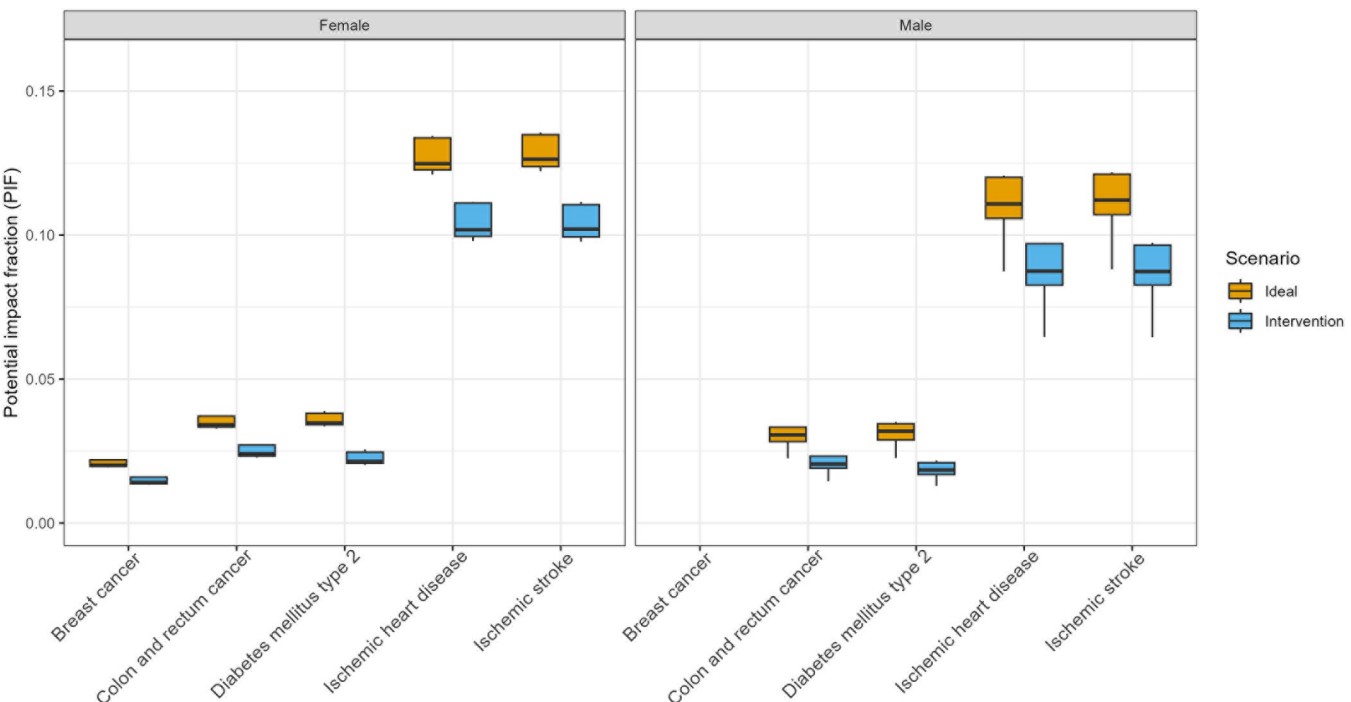

**Fig 2. Potential impact fractions used in this analysis.**

regardless of the cause of death or age group, reflecting the lower level of physical activity among females in most age groups. Third, PIFs are the highest for cardiovascular causes, which influences the projections of avoidable deaths by cause at the population level.

## Avoidable mortality and disability

By applying the three categories to develop different scenarios, at baseline, the average population distribution of MET-min/week in Saudi Arabia across age groups was 83% low, 10% moderate, and 7% high. In the intervention scenario, the low, moderate, and high groups would change to 19%, 74%, and 7%, respectively. In the ideal scenario, the low, moderate, and high groups would change to 5%, 65%, and 30%, respectively.

Overall, it is estimated that between 2023 and 2040, about 80,000 (intervention scenario) to 110,000 (ideal scenario) deaths from all causes could be avoided by increasing physical activity levels in Saudi Arabia. Fig 3 shows the annual number of deaths that could be avoided. This number would increase by about one-third between 2023 and 2040 due to population growth and aging (as would the total number of deaths in the baseline scenario).

The total number of DALYs that could be avoided ranged from 2.0 million (intervention scenario) to 2.9 million (ideal scenario). Over time, trends in avoided DALYs would be similar to those in deaths avoided shown in Fig 3, increasing by about one-third between 2023 and 2040. Fig 4A illustrates the cumulative avoidable deaths in the two scenarios and disaggregates these deaths by cause. Fig 4B illustrates the cumulative DALYs averted in the two scenarios. The overwhelming majority of deaths and DALYs would be due to ischemic heart disease, as

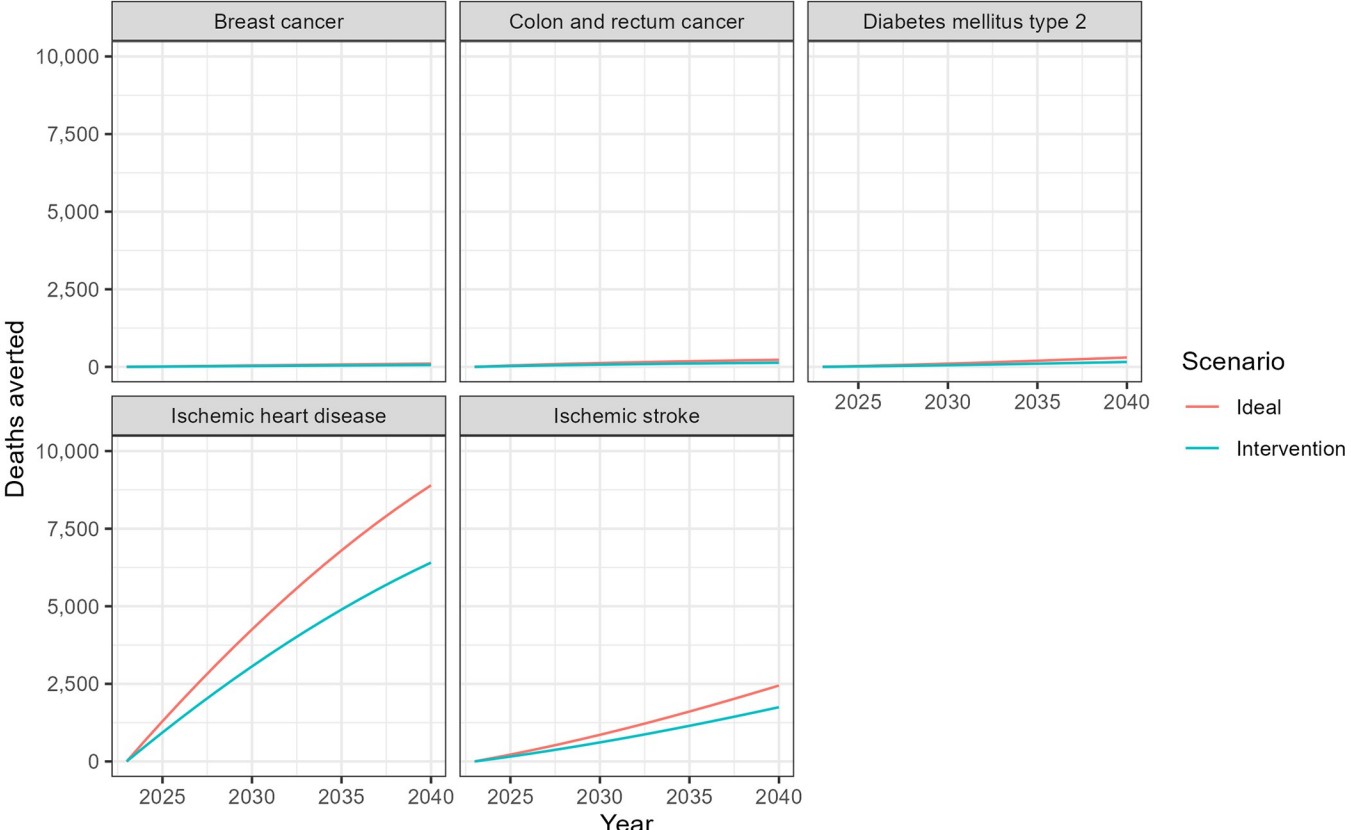

**Fig 3. Deaths from selected causes that could be avoided between 2023 and 2040 in two scenarios of increased physical activity in Saudi Arabia.**

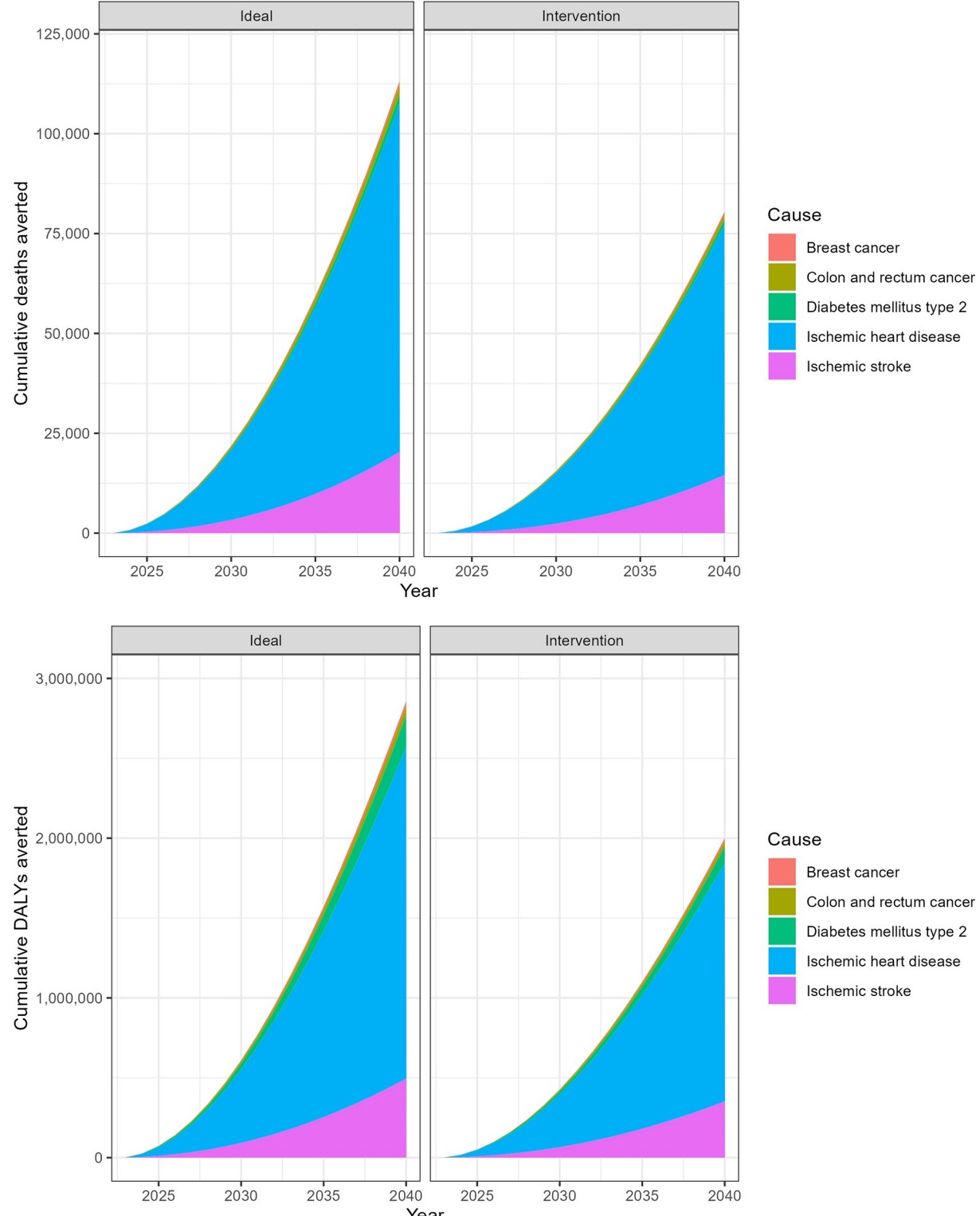

**Fig 4. a.** Cumulative cause-specific deaths avoidable between 2023 and 2040 in two scenarios of increased physical activity in Saudi Arabia. **b.** Cumulative DALYs avoidable between 2023 and 2040 in two scenarios of increased physical activity in Saudi Arabia.

**Table 1. Economic impact of insufficient physical activity in Saudi Arabia (2023–2040).** Costs are in 2022 United States dollars. Disaggregated numbers may not round up to totals due to rounding to two significant digits. DALYs = disability-adjusted life-years.

| Group | Intervention scenario | | | Ideal scenario | | |
|---|---|---|---|---|---|---|
| | *Avoidable deaths (thousands)* | *Avoidable DALYs (thousands)* | *Economic impact (US$ billions)* | *Avoidable deaths (thousands)* | *Avoidable DALYs (thousands)* | *Economic impact (US$ billions)* |
| Total | 80 | 2000 | 92 | 110 | 2900 | 130 |
| By age | | | | | | |
| *<40 yr* | 1.7 | 85 | 3.9 | 2.4 | 120 | 5.8 |
| *40–69 yr* | 61 | 1700 | 78 | 86 | 2400 | 110 |
| *70+ yr* | 18 | 240 | 11 | 25 | 330 | 15 |
| By sex | | | | | | |
| *Females* | 32 | 830 | 39 | 43 | 1200 | 54 |
| *Males* | 49 | 1200 | 54 | 70 | 1700 | 78 |

this cause has the highest PIFs (Fig 2). Conversely, the burden of breast cancer and colorectal cancer would not be significantly reduced with increased physical activity because these conditions are less common and may involve other factors, such as genetics, and their association with physical activity is lower as compared to cardiovascular diseases and diabetes.

## Economic impact

Based on the estimates of avoidable deaths and DALYs, we calculate the economic cost of avoidable disease burden using the intrinsic value approach, with the implied economic value of an avoidable DALY in Saudi Arabia being 2.3 times GDP per capita [21]. Table 1 presents the economic impact of insufficient physical activity, with results disaggregated by age, sex, and scenario. The cumulative impact ranges from US$92 billion (intervention scenario) to US$130 billion (ideal scenario), an average of US$5.4 billion to US$7.6 billion annually over the next 17 years, or 0.49% to 0.68% of 2022 GDP in Saudi Arabia.

Most of the projected economic losses of insufficient physical activity are attributable to individuals with ischemic heart disease with males being disproportionately affected. This accounts for about 60% of health and economic losses. Under the intervention scenario, approximately US$54 billion could be saved from improved physical activity in males, approximately 75% of which can be attributed to ischemic heart disease. Under the ideal scenario, US$78 billion would be saved from insufficient physical activity in males, approximately $60 billion of which would be saved due to ischemic heart disease (Table 1).

## Discussion

Overall, we estimate that between 2023 and 2040, 80,000 (intervention scenario) to 110,000 (ideal scenario) deaths from all causes could be avoided by increasing physical activity levels in Saudi Arabia. If these scenarios are not implemented, the economic value of this excess mortality and disability would be between US$92 billion and US$130 billion, respectively. Results show that most of the economic impact of insufficient physical activity would be from individuals with ischemic heart disease, and males would be disproportionately affected. The intervention scenario used in the study is intended to be a more realistic level that Saudi Arabia could achieve through governmental policies. The reason for choosing a high physical activity level for the ideal scenario was to provide an "upper bound" on the health and economic burden of insufficient physical activity.

The study findings suggest a larger economic impact of physical inactivity than that implied by previous modeling studies in the country. For example, a 2018 World Health Organization's study estimated that economic losses from all NCDs—irrespective of the attribution to physical inactivity—were about 2.8% of GDP in Saudi Arabia [26]. An update of this research on direct medical costs and worker productivity found that NCD-related losses were approximately 4.5% of GDP [27]. We found that the economic impact of physical inactivity-related NCDs alone—about 5% of the NCD burden—would be valued at around 0.5–0.7% of GDP. A reason our estimates are lower than the World Health Organization's study is that we defined the counterfactual differently: not according to the theoretical minimum distribution of physical inactivity, but according to levels seen in high-performing countries, which are still considerably higher than the theoretical minimum but reflect what is achievable through policy change. Still, insufficient physical activity is only a small part of the NCD puzzle, and failing to act on NCDs more generally would lead to very large economic losses. For example, applying the intrinsic value approach to all NCD deaths in 2019 would yield a value of around 25% of GDP globally and approximately 10% of GDP in Saudi Arabia [19]. We believe that our figures and approach best correspond to reality, though we note that our valuation approach (i.e., using the intrinsic value of health) is somewhat contested in the literature. Still, our approach is consistent with a welfare perspective that includes non-market losses [11].

This study's findings underscore an urgent need to develop innovative programs and policies to encourage physical activity, specifically to tackle cardiovascular diseases and diabetes. The main challenge with tackling insufficient physical activity, dietary risks, or both is the paucity of evidence-informed policy options. The Disease Control Priorities project recently provided recommendations for intersectoral policy action to reduce health-related risks [21]. Large-scale built-environment interventions to promote physical activity are more likely to succeed than small-scale community-based health promotion interventions, but neither has a particularly robust evidence base. The situation in Saudi Arabia is not unique; as it ranked the 3rd among 172 countries with the greatest cardiovascular disease mortality risk attributed to physical activity [2], other developed countries face similar policy challenges. Governmental agencies should consider "experimenting" with different urban planning, transportation, and infrastructure-related policies that can incentivize physical activity. They should prospectively evaluate these policies using rigorous scientific methods. Ambitious policy experimentation could allow these governments to become international examples in obesity and NCD prevention.

The distinction between the different NCD risk factors of insufficient physical activity and high body mass index (i.e., overweight and obesity) must also be emphasized. Obesity is responsible for a greater share of deaths than insufficient physical activity (22% vs. 4.8%, respectively) [3]. Hence, while insufficient physical activity is a risk factor for obesity, dietary risks appear to be much more important drivers of obesity-related disease burden in Saudi Arabia [28]. Efforts by the government and other stakeholders to tackle obesity should take a comprehensive approach, with a relatively greater emphasis on dietary risks. Unfortunately, there are no simple policy solutions to dietary risks; most World Health Organization-recommended "cost-effective" interventions have a small impact and are cost-effective merely because they are cheap [29]. Further research is needed to understand the dietary drivers of obesity and the policy interventions that can most effectively promote a healthy weight.

This study is the first to present critical findings on the cost of physical inactivity in Saudi Arabia. It enhances our understanding of the economic repercussions of insufficient physical activity in the country, addressing a substantial public health issue and providing a case for investment. The study offers a thorough assessment using both mortality and DALYs. Finally, by translating health improvements into economic returns through standard cost-benefit

analysis methods, the study provides a compelling argument for investment and practical insights for policymaking that can advance the physical activity policy agenda in Saudi Arabia.

Despite the study's importance, certain limitations warrant consideration. First, as implied previously in the discussion, the "costs" that we report here are economic costs (based on estimates of welfare losses) rather than financial outlays. Excess healthcare costs can be a more relevant measure to some stakeholders and represent "costs" borne by payers. It was beyond the scope of this article to compare multiple approaches to evaluating health losses, but future projects with sufficient data could allow for these analyses. Second, our model used population-level data and average values based on triangulating surveys and other data sources. Additionally, in order to ensure a focused and attainable research endeavor, we had to limit the scope exclusively to the four conditions exhibiting the highest burden associated with insufficient physical activity, as identified through the GBD estimates. Finally, we could not account for joint distributions of various risks (e.g., low physical activity, tobacco use, and high cholesterol) concentrated in the same high-risk individuals. An individual-level simulation model could incorporate these factors, but high-quality, individual-level data in Saudi Arabia is lacking. Population-based cohort studies of NCD risk factors and long-term outcomes would enhance local understanding of trends in major diseases and improve the accuracy and precision of modeling analyses like this study.

## Conclusions

If no additional action is taken, insufficient physical activity could lead to an excess of 80,000 to 110,000 deaths and 2.0 million to 2.9 million DALYs in Saudi Arabia between 2023 and 2040. The economic value of these health losses could be as high as 0.7% of the country's GDP. Tackling physical inactivity in Saudi Arabia will require multisectoral approaches that include redesigning transportation systems, strengthening school-based physical activity programs, behavioral and social approaches, including community clubs and classes, and effective urban planning. Given the lack of robust evidence on interventions to improve physical activity, there is an opportunity for governmental agencies to experiment with policies and rigorously evaluate their effectiveness and costs, significantly contributing to the international evidence base.

## Acknowledgments

This work was conducted by the King Faisal Specialist Hospital and Research Center (KFSHRC, Riyadh, Saudi Arabia) with technical support from the World Bank. The authors are grateful for the overall support provided by Michele Gragnolati, World Bank Practice Manager, Health Nutrition and Population, Middle East and North Africa region; Safaa El Tayeb El-Kogali, World Bank Regional Director for the GCC countries; Rekha Menon, former World Bank Practice Manager, Health Nutrition and Population, Middle East and North Africa region; and Issam Abousleiman, former World Bank Regional Director for the GCC countries.

## Author Contributions

**Conceptualization:** Saleh A. Alqahtani, David Watkins, Severin Rakic, Christopher H. Herbst, Hazzaa M. Al-Hazzaa.

**Data curation:** Amal AlGhammas, Fadiah Alkhattabi.

**Formal analysis:** David Watkins, William Msemburi, Sarah Pickersgill.

**Funding acquisition:** Saleh A. Alqahtani, Christopher H. Herbst.

**Investigation:** Mariam M. Hamza, William Msemburi, Sarah Pickersgill.

**Methodology:** Mariam M. Hamza, David Watkins, Hazzaa M. Al-Hazzaa.

**Project administration:** Reem AlAhmed, Severin Rakic.

**Resources:** Reem AlAhmed, Reem F. Alsukait.

**Software:** David Watkins, Sarah Pickersgill.

**Supervision:** Saleh A. Alqahtani, Christopher H. Herbst.

**Validation:** Saleh A. Alessy, Ada Alqunaibet, Amal AlGhammas, Fadiah Alkhattabi.

**Visualization:** William Msemburi, Sarah Pickersgill.

**Writing – original draft:** Mariam M. Hamza, David Watkins, Sarah Pickersgill, Severin Rakic.

**Writing – review & editing:** Saleh A. Alqahtani, Reem AlAhmed, Fadiah Alkhattabi, Reem F. Alsukait, Christopher H. Herbst, Hazzaa M. Al-Hazzaa.

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
