## [Decision Letter · Decision Letter 0]

12 Oct 2023

PONE-D-23-28912Health and Economic Burden of Insufficient Physical Activity in Saudi ArabiaPLOS ONE

Dear Dr. Alqahtani,

Thank you for submitting your manuscript to PLOS ONE. After careful consideration, we feel that it has merit but does not fully meet PLOS ONE’s publication criteria as it currently stands. Therefore, we invite you to submit a revised version of the manuscript that addresses the points raised during the review process.

We look forward to receiving your revised manuscript.

Kind regards,

Seyed Aria Nejadghaderi

Academic Editor

PLOS ONE

Journal Requirements:

Reviewers' comments:

Reviewer's Responses to Questions

**Comments to the Author**

1. Is the manuscript technically sound, and do the data support the conclusions?

Reviewer #1: Partly

Reviewer #2: Yes

2. Has the statistical analysis been performed appropriately and rigorously? 

Reviewer #1: Yes

Reviewer #2: Yes

3. Have the authors made all data underlying the findings in their manuscript fully available?

Reviewer #1: No

Reviewer #2: No

4. Is the manuscript presented in an intelligible fashion and written in standard English?

Reviewer #1: Yes

Reviewer #2: Yes

5. Review Comments to the Author

Reviewer #1: The authors of this study investigated the health and economic burden of insufficient physical activity in Saudi Arabia, a country located in the Middle East region where most of the countries located in this region suffer from this public health issue. The study also benefits from a projection of the findings to 2040 and the expected burden is estimated in three scenarios of baseline, intervention, and ideal which could be interesting for future public health policy making in this country. Overall, the study benefits from a robust design and statistical analysis, and the manuscript is well drafted. However, some parts of the manuscript need some revisions to enhance the quality of material, with most of the comments on the methods section. Below, please find my comments and suggestions in this regard.

1. Methods: the two major parts of the methods section including “Estimating Avoidable Health Loss” and “Estimating Economic Impact” which are the most important parts of this paper are drafted inadequately in description and authors only mentioned they have used previously developed methods. Specifically, the last part on the economic burden estimation is too incomplete and need further expansion to make this research a valid and reliable one. In this regard, the incorporated models to estimate the economic burden should be fully elaborated in this section.

2. Methods: the references for the pooled and used relative risks (RRs) to calculate the PAFs is not mentioned in this section which needs a revision to make everything is clear. Also, adding a table as a supplementary and providing the values of used RRs in this study is essential.

3. Results: the presented section on findings lack appropriate tables including the estimated numbers of health and economic burden in different stratifications of population characteristics and according to the four health conditions. Therefore, enriching this section with informative tables is highly needed.

4. Limitations: one of the limitations which is not mentioned in the last part of the discussion is that authors have made their selection of the four conditions with highest burden linked to insufficient physical activity based on the GBD estimations and this could be biased as they secondarily use another data. So, mentioning this issue is highly suggested.

Reviewer #2: 1- Be consistent in using abbreviations. You should define abbreviations in their first use. Moreover, if you have defined any abbreviations, you should only use them instead of full terms (e.g., you used PA and “physical activity” many times interchangeably)

2- In the last paragraph of introduction, you should use present or future verbs. This should explain the main aim of your study.

3- Add your study strengths prior to the limitations section.

4- Revise the whole manuscript for typos and grammatical errors.

6. PLOS authors have the option to publish the peer review history of their article (what does this mean?). If published, this will include your full peer review and any attached files.

Reviewer #1: **Yes: **Sina Azadnajafabad, MD, MPH

Reviewer #2: No

---

## [Author Response · Author response to Decision Letter 0]

27 Dec 2023

We have revised the manuscript and provided point-to-point response in a separate document. Thanks

---

## [Decision Letter · Decision Letter 1]

2 Jan 2024

Health and Economic Burden of Insufficient Physical Activity in Saudi Arabia

PONE-D-23-28912R1

Dear Dr. Alqahtani,

We’re pleased to inform you that your manuscript has been judged scientifically suitable for publication and will be formally accepted for publication once it meets all outstanding technical requirements.

Kind regards,

Seyed Aria Nejadghaderi

Academic Editor

PLOS ONE

Additional Editor Comments (optional):

Reviewers' comments:

Reviewer's Responses to Questions

**Comments to the Author**

1. If the authors have adequately addressed your comments raised in a previous round of review and you feel that this manuscript is now acceptable for publication, you may indicate that here to bypass the “Comments to the Author” section, enter your conflict of interest statement in the “Confidential to Editor” section, and submit your "Accept" recommendation.

Reviewer #1: All comments have been addressed

Reviewer #2: All comments have been addressed

2. Is the manuscript technically sound, and do the data support the conclusions?

Reviewer #1: Yes

Reviewer #2: (No Response)

3. Has the statistical analysis been performed appropriately and rigorously? 

Reviewer #1: Yes

Reviewer #2: (No Response)

4. Have the authors made all data underlying the findings in their manuscript fully available?

Reviewer #1: Yes

Reviewer #2: (No Response)

5. Is the manuscript presented in an intelligible fashion and written in standard English?

Reviewer #1: Yes

Reviewer #2: (No Response)

6. Review Comments to the Author

Reviewer #1: With many thanks for the amendments and revisions, the current manuscript has addressed the raised issues. I have no further comments or suggestions.

Reviewer #2: The authors have addressed the comments properly. I have no further comments. Congratulations for this work.

7. PLOS authors have the option to publish the peer review history of their article (what does this mean?). If published, this will include your full peer review and any attached files.

Reviewer #1: **Yes: **Sina Azadnajafabad, MD, MPH

Reviewer #2: No

---

## [Editor Report · Acceptance letter]

23 Jan 2024

PONE-D-23-28912R1 

PLOS ONE

Dear Dr. Alqahtani, 

I'm pleased to inform you that your manuscript has been deemed suitable for publication in PLOS ONE. Congratulations! Your manuscript is now being handed over to our production team.

Kind regards, 

on behalf of

Dr. Seyed Aria Nejadghaderi 

Academic Editor

PLOS ONE